# Inhibition of Xanthine Oxidase Protects against Diabetic Kidney Disease through the Amelioration of Oxidative Stress via VEGF/VEGFR Axis and NOX-FoxO3a-eNOS Signaling Pathway

**DOI:** 10.3390/ijms24043807

**Published:** 2023-02-14

**Authors:** Keum-Jin Yang, Won Jung Choi, Yoon-Kyung Chang, Cheol Whee Park, Suk Young Kim, Yu Ah Hong

**Affiliations:** 1Clinical Research Institute, Daejeon St. Mary’s Hospital, 64, Daeheung-ro, Jung-gu, Daejeon 34943, Republic of Korea; 2Division of Nephrology, Department of Internal Medicine, College of Medicine, The Catholic University of Korea, 222, Banpo-daero, Seocho-gu, Seoul 06591, Republic of Korea

**Keywords:** diabetic kidney disease, xanthine oxidase, oxidative stress, VEGF, NADPH oxidase

## Abstract

Xanthine oxidase (XO) is an important source of reactive oxygen species. This study investigated whether XO inhibition exerts renoprotective effects by inhibiting vascular endothelial growth factor (VEGF) and NADPH oxidase (NOX) in diabetic kidney disease (DKD). Febuxostat (5 mg/kg) was administered to streptozotocin (STZ)-treated 8-week-old male C57BL/6 mice via intraperitoneal injection for 8 weeks. The cytoprotective effects, its mechanism of XO inhibition, and usage of high-glucose (HG)-treated cultured human glomerular endothelial cells (GECs) were also investigated. Serum cystatin C, urine albumin/creatinine ratio, and mesangial area expansion were significantly improved in febuxostat-treated DKD mice. Febuxostat reduced serum uric acid, kidney XO levels, and xanthine dehydrogenase levels. Febuxostat suppressed the expression of VEGF mRNA, VEGF receptor (VEGFR)1 and VEGFR3, NOX1, NOX2, and NOX4, and mRNA levels of their catalytic subunits. Febuxostat caused downregulation of Akt phosphorylation, followed by the enhancement of dephosphorylation of transcription factor forkhead box O3a (FoxO3a) and the activation of endothelial nitric oxide synthase (eNOS). In an in vitro study, the antioxidant effects of febuxostat were abolished by a blockade of VEGFR1 or VEGFR3 via NOX-FoxO3a-eNOS signaling in HG-treated cultured human GECs. XO inhibition attenuated DKD by ameliorating oxidative stress through the inhibition of the VEGF/VEGFR axis. This was associated with NOX-FoxO3a-eNOS signaling.

## 1. Introduction

The global prevalence of diabetes mellitus has been steadily increasing. Diabetic kidney disease (DKD) is currently the leading cause of renal failure and is a significant burden on public health [1,2]. The mainstay therapy for DKD is early detection and strict glucose and blood pressure control via renin–angiotensin system blockade [3]. Nevertheless, the need for innovative treatments to prevent or slow the progression of DKD remains unmet. Although molecular mechanisms involved in the pathogenesis of DKD are extremely complex and have yet to be fully elucidated, oxidative stress—which is caused by excessive reactive oxygen species (ROS) surpassing existing antioxidative defense mechanisms—plays a crucial role in the pathogenesis of DKD [4].

Uric acid, the final product of purine metabolism, has both oxidant and antioxidant properties due to ROS production. Xanthine oxidoreductase (XOR) is an enzyme that catalyzes the sequential hydroxylation of hypoxanthine to xanthine to uric acid by utilizing either NAD^+^ or O_2_, with the accompanying production of ROS. XOR appears in two interconvertible forms, xanthine dehydrogenase (XDH) and xanthine oxidase (XO), and is initially synthesized from XDH, which can rapidly be converted into XO [5]. XO delivers electrons directly to O_2_, thus generating two ROS: superoxide anion (O_2_^•−^) and hydrogen peroxide (H_2_O_2_) [5]. Increased XO activity accounts for significant high glucose (HG)-induced ROS production in diabetes [6]. In previous studies, reducing the serum uric acid level was shown to have protective effects against the progression of DKD in type 1 and type 2 diabetic mouse models [7,8]. Thus, understanding XO inhibition based on the production of ROS may help with the development of novel therapeutic targets for DKD.

HG-induced ROS production plays a substantial role in endothelial cell senescence [9], and glomerular endothelial cell dysfunction plays a crucial role in the development and progression of DKD [10]. Circulating XOR changes to an oxidase form and binds to endothelial cells, and XOR-derived ROS affects the microvascular lining by inducing endothelium permeabilization and endothelial cell injury [5]. Glomerular endothelial cells (GECs) may contribute to the progression of DKD through crosstalk with other cells in the glomerulus (podocytes and mesangial cells), while vascular endothelial growth factors (VEGFs) are the major mediators for GEC and podocyte communication [10]. Diabetic mouse models showed increased renal VEGF expression, and VEGF was implicated in the pathogenesis of DKD [11,12].

A potent non-purine selective XO inhibitor, febuxostat, is often used to treat hyperuricemia in clinical practice [13]. Previous studies have reported that XO inhibitors protect against DKD in type 1 and type 2 diabetic mouse models, but they did not elucidate the precise mechanism of XO inhibition for renoprotective effects against DKD [8,14,15,16,17]. A recent study demonstrated that XO inhibition protected against endometrial hyperplasia by improving oxidative stress and inflammation by improving uterine-reduced glutathione (GSH) and superoxide dismutase (SOD) and inhibiting the expressions of phosphatidylinositol-3-kinase (PI3K), Akt and VEGF [18]. Therefore, this study aimed to investigate whether an XO inhibitor could attenuate oxidative stress and protect against DKD by inhibiting VEGF-NADPH oxidase (NOX) signaling pathway in an animal model of DKD and human GECs.

## 2. Results

### 2.1. The Effects of XO Inhibition on Physical and Biochemical Parameters in STZ-Induced DKD Mice

The physical and biochemical parameters as of the end of the experimental period are presented in Table 1 and Figure 1. There was a significant decrease in body weight, while food and water intake increased, but without significant difference in the streptozotocin (STZ) and STZ + Feb groups compared with the Cont group. The kidney weight/body weight ratio significantly increased in the STZ group compared with the Cont and Feb groups and significantly decreased in the STZ + Feb group (Table 1, *p* < 0.01). Fasting blood glucose and HbA1c levels in the STZ and STZ + Feb groups significantly increased compared with the Cont and Feb groups but had a non-significant difference between the two groups. Therefore, febuxostat treatment per se did not appear to have any effect on lowering blood glucose (Figure 1 and Table 1). Although the serum creatinine levels were not significantly different between the four groups, urine albumin-to-creatinine ratio (ACR) and serum cystatin C levels significantly increased in the STZ group, and these parameters were improved in the STZ + Feb group (*p* < 0.05 and *p* < 0.001, respectively; Figure 1b,c).

### 2.2. The Effects of XO Inhibition on Uric Acid and XOR Activities in STZ-Induced DKD Mice

The serum uric acid level did not change in the STZ group, while the increase in uric acid was significantly suppressed in the STZ + Feb group (*p* < 0.05, Figure 1d). Kidney XO and XDH activities were significantly increased in the STZ group and were reduced to baseline in the STZ + Feb group (*p* < 0.01 and *p* < 0.001, respectively; Figure 1e,f).

**Table 1 ijms-24-03807-t001:** Physical and biochemical parameters of the four groups at the end of the 8-week experimental period.

	Cont	Feb	STZ	STZ + Feb
BW (g)	29.6 ± 0.8	27.0 ± 2.1	22.2 ± 2.0 ***^,††^	22.6 ± 2.6 ***^,††^
Food intake (g/day)	3.86 ± 0.20	3.20 ± 0.33	5.60 ± 0.13 ***^,†††^	6.17 ± 0.38 ***^,†††^
Water intake (mL/day)	6.33 ± 1.04	4.53 ± 0.23	23.4 ± 0.88 ***^,†††^	24.2 ± 2.46 ***^,†††^
KW/BW(g/g × 100)	0.7 ± 0.07	0.6 ± 0.06	0.9 ± 0.02 *^,††^	0.7 ± 0.1 ^‡^
FBS (mg/dL)	180.0 ± 22.3	172.8 ± 25.3	681.0 ± 155.2 ***	489.9 ± 39.0 ***^,††^
HbA1c (%)	4.23 ± 0.31	4.16 ± 0.32	11.3 ± 1.32 ***^,†††^	10.1 ± 1.48 ***^,†††^
Serum Cr (mg/dL)	0.08 ± 0.05	0.08 ± 0.04	0.07 ± 0.05	0.04 ± 0.05

Abbreviations: STZ—streptozotocin; KW—kidney weight; BW—body weight; FBS—fasting blood glucose; HbA1c—Hemoglobin A1c; Cr—creatinine. * *p* < 0.05, and *** *p* < 0.001 compared with Cont groups, ^††^ *p* < 0.01 and ^†††^ *p* < 0.001 compared with Feb groups, ^‡^
*p* < 0.05 compared with STZ groups.

**Figure 1 ijms-24-03807-f001:**
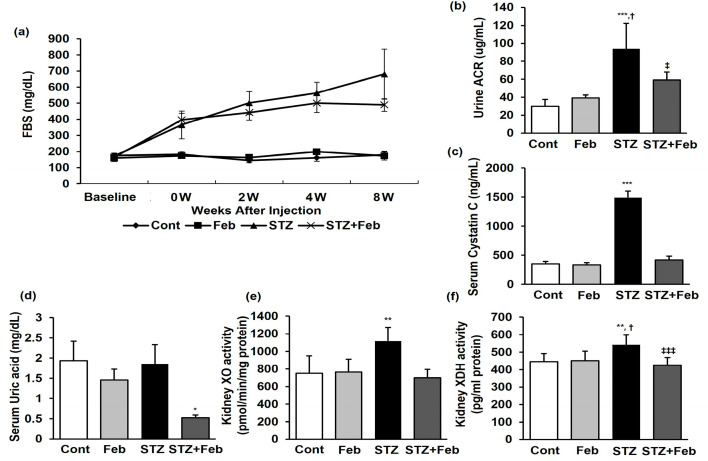
The changes in glucose, renal function, and XOR activities in STZ-induced diabetic mice with or without febuxostat treatment. (**a**) Fasting blood glucose, (**b**) Urine Albumin/Creatinine ratio, *** *p* < 0.001 vs. Cont, ^†^ *p* < 0.05 vs. Feb, and ^‡^ *p* < 0.05 vs. STZ group. (**c**) Serum cystatin C, *** *p* < 0.001 vs. other groups, (**d**) serum uric acid, * *p* < 0.05 vs. other groups, (**e**) kidney XO activity, ** *p* < 0.01 vs. other groups, (**f**) kidney XDH activity, ** *p* < 0.01 vs. Cont, ^†^ *p* < 0.05 vs. Feb, and ^‡‡‡^ *p* < 0.001 vs. STZ group.

### 2.3. The Effects of XO Inhibition on Renal Histological Changes, and Expression of TGF-β1 and Col IV in STZ-Induced DKD Mice

The fractional mesangial area was not different between the Cont and Feb groups, and further increased in the STZ group (Figure 2, *p* < 0.001). The expression of transforming growth factor-β1 (TGF-β1) and type IV collagen (Col IV) was also significantly increased in the STZ group compared with the Cont and Feb groups (Figure 2, *p* < 0.001 and *p* < 0.001, respectively). All diabetes-induced mesangial matrix expansion and expression of TGF-β1 and Col IV observed in the STZ group was recovered in the STZ + Feb group.

### 2.4. The Effects of XO Inhibition on VEGF and VEGFR Expression in STZ-Induced DKD Mice

The renal expression of *VEGF* mRNA was substantially increased in the STZ group and decreased in the STZ + Feb group (Figure 3a, *p* < 0.001). In addition, the changes in renal expression of VEGFR family members, VEGFR1, VEGFR2, and VEGFR3, in the STZ + Feb group were examined. The expression of VEGFR1, VEGFR2, and VEGFR3 increased in the STZ group compared with the Cont and Feb groups. VEGFR1 and VEGFR3 expression was significantly decreased in the STZ + Feb group (Figure 3b,c,e, *p* < 0.05 and *p* < 0.05, respectively) but VEGFR2 expression was unchanged in the STZ group (Figure 3d).

### 2.5. The Effects of XO Inhibition on NOX Expressions in STZ-Induced DKD Mice

Next, intrarenal NOX1, NOX2, and NOX4 expressions were measured to investigate the antioxidative effects associated with NOX signaling by febuxostat in diabetic mice. NOX1, NOX2, and NOX4 expressions were significantly enhanced in the STZ group, and the enhancement of NOX expression caused by STZ was significantly decreased in the STZ + Feb group (Figure 4a–d, *p* < 0.001, *p* < 0.001, and *p* < 0.01, respectively). Among NOX subunits, the expressions of *NoxO1, p22phox, p47phox,* and *p67phox* mRNA in the mouse kidney were significantly increased in the STZ group. The increased expressions of NOX subunits caused by STZ administration were significantly reduced in the STZ + Feb group (Figure 4e–h).

### 2.6. The Effects of XO Inhibition on Akt, FoxOs, and eNOS Expression in STZ-Induced DKD Mice

Intrarenal phospho-Ser^473^ Akt, phospho-Ser^256^ forkhead box O transcription factor (FoxO)1, phospho-Ser^253^ FoxO3a, and phospho-Ser^1173^ eNOS levels were determined using immunoblot analysis. The phospho-Ser^473^ Akt/total Akt ratio was enhanced in the STZ group and significantly inhibited in the STZ + Feb group (Figure 5a,b, *p* < 0.05). The ratios of phospho-Ser^256^ FoxO1/total FoxO1 and phospho-Ser^253^ FoxO3a/total FoxO3a were significantly increased in the kidneys of mice in the STZ group compared with mice in the Cont and the Feb groups (Figure 5a and Figure 5c,d, *p* < 0.05 and *p* < 0.05, respectively). FoxO3a phosphorylation was significantly decreased in the kidneys of mice in the STZ + Feb group (Figure 5a,d, *p* < 0.001), but FoxO1 phosphorylation was unchanged. The phospho-Ser^1173^ eNOS/total eNOS ratio was increased in the STZ group and further increased in the STZ + Feb group (Figure 5a,e, *p* < 0.01).

### 2.7. The Effects of XO Inhibition on Oxidative Stress in STZ-Induced DKD Mice

Malondialdehyde (MDA) level in the kidneys of mice in the STZ group significantly increased compared with that of with mice in the Cont and Feb groups, whereas it significantly decreased in mice in the STZ + Feb group (Figure 6a, *p* < 0.001). Superoxidase dismutase (SOD) activity significantly increased in the Feb group but significantly decreased in the STZ group and was significantly increased in the STZ + Feb group (Figure 6b, *p* < 0.01). In addition, SOD1 expression significantly decreased in the STZ group and substantially restored in the STZ + Feb group (Figure 6c, *p* < 0.01). In contrast, SOD2 expression did not change among the experimental groups. The immunohistochemical staining of 8-hydroxy-2′-deoxyguanosine (8-OH-dG) was increased in the STZ group, and 8-OH-dG was substantially decreased in the STZ + Feb group, reflecting an amelioration of oxidative stress in the kidney tissues of diabetic mice (Figure 6d, *p* < 0.001).

### 2.8. The Effects of XO Inhibition on Oxidative Stress in HG-Treated Human GECs via VEGFR1/3-Dependent NOX-FoxO3a-eNOS Signaling

The effects of XO inhibition on HG-induced oxidative stress associated with VEGFR inhibition were investigated using VEGFR1 and VEGFR3 inhibitors in cultured human GECs. The expression of NOX1, NOX2, and NOX4 was significantly increased by HG treatment and significantly inhibited with febuxostat treatment in HG-treated human GECs. Despite febuxostat treatment, the expression of NOX1, NOX2, and NOX4 in HG-treated human GECs with anti-Flt1 peptide or SAR131675 was significantly enhanced compared with HG-treated human GECs without anti-Flt1 peptide or SAR131675 (Figure 7a–d). Consistent with changes in NOXs expression, the FoxO3a phosphorylation expression significantly increased as a result of HG treatment and was markedly inhibited following febuxostat treatment in HG-treated human GECs. The FoxO3a phosphorylation was significantly increased in HG-treated human GECs with anti-Flt1 peptide or SAR131675 despite febuxostat treatment (Figure 7a,e). The eNOS phosphorylation was significantly increased in HG treatment and further enhanced with febuxostat treatment in HG-treated human GECs. The eNOS phosphorylation was significantly decreased in HG-treated human GECs with anti-Flt1 peptide or SAR131675 despite febuxostat treatment (Figure 7a,f). In addition, HG treatment increased the number of DCF-DA-positive cells, and febuxostat treatment significantly decreased the intracellular ROS level measured based on DCF-DA in HG-treated human GECs. Anti-Flt1 peptide or SAR131675 treatment abolished these antioxidant effects caused by febuxostat treatment in HG-treated human GECs (Figure 8, *p* < 0.001).

## 3. Discussion

The results of this study provided experimental evidence that STZ-induced DKD was accompanied by increased XO, which was associated with increased VEGF/VEGFR1 and VEGFR3 levels in the kidneys. These changes led to the activation of NOX1, NOX2, and NOX4 expression, followed by FoxO3a phosphorylation, which subsequently resulted in increased oxidative stress. The non-purine selective XO inhibitor febuxostat suppressed VEGF/VEGFR1 and VEGFR3 and subsequently decreased NOXs expressions and catalytic subunits of NOXs. The protective effects of XO inhibition for DKD were attributed to the dephosphorylation of Akt and FoxO3a and the enhancement of eNOS phosphorylation, which reversed renal oxidative stress. To the best of our knowledge, this is the first study in which the hypothesis for renoprotective mechanisms of XO inhibition in type 1 DKD through VEGF inhibition by suppressing the NOX-FoxO3a-eNOS signaling pathway was elucidated.

XO is a major source of ROS in various physiologic and pathologic conditions [19]. In previous studies, XO activation was suggested to play a vital role in the pathogenesis of DKD. Liu et al. demonstrated that serum uric acid and urinary allantoic acid levels were positively correlated with albuminuria in STZ-induced diabetic rats [20]. Furthermore, the authors showed that activated XO associated with enhanced intracellular ROS caused renal damage in type 1 DKD [20]. Other studies have highlighted increased XOR activity and XO-derived oxidative stress in the kidneys of type 1 or type 2 DKD animal models [8,14,17]. In the present study, XO and XDH activities in the kidneys were significantly increased in STZ-induced DKD mice, and these findings were accompanied with the increase in albuminuria and serum cystatin C levels and the aggravation of glomerular injury. In addition, STZ-induced DKD mice showed elevated oxidative stress markers, including MDA and 8-OH-dG, in kidney tissues. Two previous findings supported our results. The levels of 8-OHdG, MDA, and H_2_O_2_ in the Graves’ ophthalmopathy orbital fibroblasts were higher than those in the normal controls [21]. Additionally, MDA and H_2_O_2_ increased in common carp liver oxidative damage caused by toxic 4-tert-butylphenol [22]. Our results confirmed that XO-derived oxidative stress may contribute to the development and progression of DKD by enhancing ROS.

Some researchers have suggested that hyperglycemia-related endothelial cell injury may be crucial in the pathological process of glomerular endothelial dysfunction in DKD [23]. Glomerular endothelial dysfunction in DKD represents the destruction of endothelial cell fenestration, increased cell proliferation, immature angiogenesis, and increased endothelial-to-mesenchymal transition [10]. Human XOR is localized in various organs including the vascular endothelium [24]. The XOR level in human plasma is usually very low, but pathologic conditions cause XOR to be released from damaged cells into the circulation, where it is converted to XO [25]. Circulating XO can bind with sulfated glycosaminoglycans on the surface of endothelial cells by competing for the binding sites of heparin [26], thus promoting endothelial activation during pro-oxidant and pro-inflammatory states, and consequently contributing to endothelial dysfunction [27,28]. Itano et al. suggested that XO inhibition attenuates glomerular endothelial damage and restores glomerular permeability in type 1 diabetic mice [17]. In addition to the findings of previous studies, we revealed the possible mechanisms responsible for the beneficial effect of XO inhibition on endothelial cell damage of DKD caused by oxidative stress in the present study using in vivo and in vitro experiments.

VEGF is a key regulator in endothelial cell survival, proliferation, and angiogenesis, and increased glomerular expression of VEGF is associated with the pathogenesis of DKD [11]. It has previously been suggested that there may be interactions between XO and VEGF in endothelial cells. Kuo et al. reported that exogenous XO induced ROS production through VEGF activation, while XO inhibition suppressed VEGF-stimulated Akt phosphorylation and reduced endothelial cell viability, proliferation, and angiogenesis via the p38 MAPK–PI3K pathway [29]. XO inhibition also showed decreased lung VEGF expression in paraquat-induced lung injury, which may be attributed to decreased oxidative stress that results from downregulation of the receptor for advanced glycation end products (RAGE)/PI3K/Akt pathway [30]. Similarly, in the present study, results confirmed the interaction between XO and VEGF expression through the suppression of *VEGF* mRNA levels and Akt phosphorylation and the upregulation of eNOS expression caused by febuxostat treatment in STZ-induced DKD mice.

The VEGF family exerts biological actions via specific binding to tyrosine kinase receptors on the cell surface [31]. The pathogenetic roles for other VEGFRs are complex and remain largely undefined in DKD. Among the three VEGFRs, VEGFR1 has a higher affinity for VEGFA than VEGFR2, but VEGFR2 exerts strong tyrosine kinase activity [31]. The interactions between VEGFA and VEGFR2 result in the activation of downstream signaling and most cellular actions of VEGF in DKD [32,33]. Because VEGFR1 is considered a negative modulatory regulator of VEGFA, numerous experiments have shown that VEGFR1 expression was decreased in DKD [34,35]. However, VEGFR1 demonstrates a dual action, playing a negative role in angiogenesis in the embryo, most likely by trapping VEGFA, while it plays a positive role in macrophage function, inflammation, and atherosclerosis in adulthood [36]. Some experiments have shown that both VEGF and VEGFR1 expression increased in HG-treated proximal tubular cells and the early stages of DKD [37,38]. Sato et al. reported that podocyte VEGF expression positively correlated with infiltration of VEGFR1-positive macrophages in diabetic eNOS KO mice, while NO negatively regulated VEGF-induced macrophage migration by inhibiting VEGFR1 expression in GECs [39]. In addition, VEGFR3 expression was increased in type 2 diabetic mice [40]. In the present study, the expression of all VEGFRs increased in STZ-induced DKD mice, and VEGFR1 and VEGFR3 expression was hindered by XO inhibition. In human GECs, HG-induced ROS production and signal transduction associated with oxidative stress and endothelial cell homeostasis were significantly decreased due to XO inhibition, and these changes were abolished with VEGFR1 or VEGFR3 inhibition. These conflicting results regarding VEGFR1 expression in DKD are difficult to explain; however, we hypothesized that XO inhibition was mediated by the VEGF/VEGFR1 and VEGFR3 axis to enable its biologic actions associated with endothelial function and oxidative stress in DKD. Further research is needed to clarify the biological functions and mechanisms of XO inhibition through the VEGF/VEGFR axis in DKD.

NOX, a major source of endothelial ROS, is activated by various stimuli, including VEGF, which stimulates ROS production via activation of Rac1-dependent NOX in endothelial cells [41]. NOX-dependent intrarenal ROS overproduction contributes to the initiation and progression of DKD [4]. Among the major NOX isoforms, NOX4 is of specific interest in DKD due to its enrichment in kidney tissue. All major catalytic subunits, NOX1, NOX2, NOX4, and NOX5, and regulatory subunits such as p22phox, p47phox, and p67phox, can be found in kidney tissues including endothelial cells [4,42]. In the current study, NOX1, NOX2, and NOX4 expression and expression of cytosolic subunits NoxO1, p22phox, p47phox, and p67phox were upregulated in STZ-induced DKD mice and HG-treated human GECs and significantly reduced as a result of febuxostat treatment. These findings indicate that the suppression of NOXs functions as a mediator to prevent disease progression by reducing oxidative stress via inhibition of XO in DKD.

FoxO transcription factors play a crucial role in regulating cellular homeostasis by inducing the expression of genes involved in cellular metabolism and resistance to oxidative stress [43]. Our previous studies revealed that diabetic conditions induced PI3K and Akt, leading to FoxO1 and FoxO3a phosphorylation in the type 2 DKD model using *db*/*db* mice [44,45]. In addition, we recently reported that XO inhibition attenuated contrast-induced acute kidney injury by modulating oxidative stress through the inhibition of NOX1 and NOX2, while simultaneously enhancing dephosphorylation of FoxO1 and FoxO3a [46]. In this study, NOX expression and FoxO1 and FoxO3a phosphorylation increased in STZ-induced DKD mice, whereas XO inhibition suppressed NOX activation and FoxO3a phosphorylation. Febuxostat-induced FoxO3a dephosphorylation and NOX suppression were diminished by VEGFR1 inhibitor or VEGFR3 inhibitor in HG-treated human GECs. Our data suggest that targeting the VEGF-NOXs-FoxO3a signaling pathway through the XO inhibition could have beneficial effects, alleviating intracellular oxidative stress in DKD.

## 4. Materials and Methods

### 4.1. Animals and Experimental Design

Male C57BL/6J mice, aged 8–10 weeks old and weighing approximately 20–22 g, were purchased from Daehan Biolink (Eumseong, Chungcheonbuk-do, Republic of Korea). Mice were randomly assigned into the following groups: control (Cont, n = 6), febuxostat-treated (Feb, n = 6), STZ-induced DKD (STZ, n = 8), and STZ-induced DKD treated with febuxostat (STZ + Feb, n = 8). To establish an animal model of DKD, 50 mg/kg of STZ/0.1 M sodium citrate buffer (pH 4.5) (Sigma-Aldrich, St. Louis, MO, USA) was injected into mice intraperitoneally after fasting for 4 h daily for 5 days. Seven days after the last STZ injection, mice with a fasting blood glucose level ≥ 250 mg/dL were selected for the study. Febuxostat (SK Chemical, Sungnam, Gyeonggi-do, Republic of Korea) was suspended in 0.5% carboxymethyl cellulose (CMC) and administered daily to mice at 5 mg/kg for 8 weeks via oral gavage. The control used for STZ was 0.1 M sodium citrate buffer and 0.5% CMC was used as the control for febuxostat. At the end of the experiments, mice were placed on a 96-well plate in a customized spot urine cage for 4 h and allowed to roam freely on the plate until spontaneous urination occurred. Urine free of fecal contamination was collected using a pipette. Mice were sacrificed 8 h after fasting, and their kidneys and blood were harvested.

### 4.2. The Assessments of Biochemical Parameters in Blood and Urine Samples

Fasting blood glucose levels were measured at 0, 2, 4, and 8 weeks after febuxostat treatment using an Accu-check meter (Roche Diagnostics, St. Louis, MO, USA). After scarifying, blood HbA1c was measured using a DCA Vantage Analyzer (Siemens Healthineers, Erlangen, Germany) and serum cystatin C level was assessed with a mouse cystatin C enzyme-linked immunosorbent assay (ELISA) kit (R&D Systems, Minneapolis, MN, USA). The serum creatinine, urine creatinine, and serum uric acid levels were checked using an IDEXX VetTest^®^ Chemistry Analyzer (IDEXX Laboratories, Inc., Westbrook, ME, USA). Urinary albumin excretion was assessed via ACR on spot urine collection. Urine albumin levels were quantified using a Mouse Albumin ELISA Quantification Set (Bethyl Laboratories, Montgomery, TX, USA), following the manufacturer’s protocols.

### 4.3. Histopathologic Analysis and Immunohistochemical Staining

To observe histological changes in kidneys, 4 µm paraffin-embedded sections were stained with periodic acid–Schiff (PAS) stain. Semi-quantification of the fractional mesangial area in the glomerulus was performed with more than 30 glomeruli per kidney in a blinded manner (original magnification 400×). To assess the severity of the mesangial matrix expansion, the fraction mesangial area was calculated as the proportion of the area of mesangial matrix to the total area of glomerulus and presented in percentage by determining the color intensity per glomerulus using Image-J software (the National Institutes of Health, Bethesda, MD, USA) [47]. Immunohistochemistry was performed with antibodies for TGF-β1 (R&D Systems, Minneapolis, MN, USA), Col IV (Biodesign International, Saco, ME, USA), and 8-OH-dG (JaICA, Nikken SEIL Co., Ltd., Shizuoka, Japan), followed by red color development according to the manufacturer’s protocols (Vectastain and Vector NovaRed HRP substrate kit; Vector Laboratories, Burlingame, CA, USA). Semi-quantification of positive signals was performed in a blind manner using a light microscope (Olympus BX-50; Olympus Optical, Tokyo, Japan).

### 4.4. In Vitro Experiments

Human GECs were obtained from the American Type Culture Collection (Manassas, VA, USA) and incubated in Dulbecco’s modified Eagle’s medium (DMEM) containing 10% fetal bovine serum, 50 U/mL penicillin, and 50 μg/mL streptomycin. Human GECs were exposed to HG with 33 mM/L D-glucose (Thermo Fisher Scientific, Seoul, Republic of Korea), or a similar volume of 27.4 mM/L D-mannitol was used as the osmotic low-glucose control (LG). The Feb group was treated with 0.5 µM of febuxostat dissolved in dimethyl sulfoxide (DMSO) and the Cont group with DMSO for 48 h after 24 h of HG or LG condition.

To evaluate the direct effect of VEGFRs on febuxostat, VEGFR inhibitors were used as follows: 0.1 mM of anti-Flt1 peptide (GNQWFI) was used as a VEGFR1 inhibitor and 10 nM of was used SAR131675 as a VEGFR3 inhibitor. Anti-Flt1 peptide (GNQWFI) or SAR131675 were treated 1 h prior to febuxostat administration in HG-treated human GECs.

### 4.5. Enzyme Immunoassay in the Kidney Tissues

The kidney tissues were homogenized in PRO-PREPTM Protein Extraction Solution (iNtRON Biotechnology, Seongnam, Republic of Korea), followed by centrifugation at 12,000× *g* at 4 °C for 10 min. The activities of XO and XDH in kidneys were determined with an XO (LSBio, Seattle, WA, USA) and XDH ELISA Kit (MyBiosource, San Diego, CA, USA), respectively. The MDA concentration, a byproduct of lipid peroxidation, was measured using OxiSelect MDA Adduct ELISA kit (Cell Biolabs Inc., San Diego, CA, USA) to assess the severity of oxidative stress. SOD activity was determined using the Superoxidase Dismutase Activity Assay Kit (Abnova, Walnut, CA, USA) following the manufacturer’s manuals. Equal amounts of kidney protein were measured using the Bradford assay (Bio-Rad Laboratories, Hercules, CA, USA). The concentration of each sample was calculated by comparing the optimal density (OD) of each sample at a wavelength of 450 nm with an OD standard curve.

### 4.6. Immunoblot Analysis

The proteins extracted from kidney tissues and human GECs were subjected to sodium dodecyl sulfate–polyacrylamide gel electrophoresis and transferred onto a nitrocellulose membrane. The membrane was incubated with primary antibodies against the following target proteins: VEGFR1 (Boster, Pleasanton, CA, USA), VEGFR2 (St John’s Laboratory, London, UK), VEGFR3 (NSJ Bioreagent Logo, San Diego, CA, USA), NOX1 and NOX4 (Lifespan Biosciences, Seattle, WA, USA), NOX2 (BD Biosciences, San Jose, CA, USA), total Akt, phospho-Ser^473^ Akt, SOD1 and SOD2 (Cell Signaling Technology, Danvers, MA, USA), total FoxO1, phospho-Ser^256^ FoxO1, total FoxO3a, and phospho-Ser^253^ FoxO3a (Novus Biologicals, Centennial, CO, USA), total endothelial nitric oxide synthase (eNOS, EMD Millipore, Middlesex County, MA, USA), and phospho-Ser^1173^ eNOS (Cell Signaling Technology, Danvers, MA, USA). Glyceraldehyde 3-phosphate dehydrogenase (GAPDH) was blotted as internal loading controls (Cell Signaling Technology, Danvers, MA, USA). The transferred proteins were visualized using a chemiluminescent detection system (Amersham Pharmacia Biotech, London, UK), and the detection of positive band density was performed by the ChemiDoc™ XRS+ System (Bio-Rad Laboratories, Hercules, CA, USA).

### 4.7. Real-Time Reverse Transcription Polymerase Chain Reaction (qRT-PCR)

Total RNA from kidney tissues was extracted using TRIzol reagent (Invitrogen, Middlesex, MA, USA). Complementary DNA was produced from messenger RNA in a reverse transcription reaction by using Reverse Transcriptase Premix (Elpis Biotech, Daejeon, Republic of Korea) and the primers (GenoTech Corporation, Daejeon, Republic of Korea) were then amplified using a Power SYBR^®^ Green polymerase chain reaction (PCR) Master Mix (Applied Biosystems, Foster City, CA, USA) with gene-specific primer pairs (Table 2). Relative amplitude of gene expression was quantified using an ABI 7500 FAST instrument and 7500 Software v2.0.6 (Applied Biosystems, Foster City, CA, USA). GAPDH was used to normalize the quantitative gene expression data.

### 4.8. Measurement of ROS in GECs

Intracellular ROS level was measured after staining with the nonfluorescent cell-permeating compound 2′-7′-dichlorofluorescein diacetate (DCF-DA) to confirm the antioxidant effects of febuxostat in HG-treated human GECs. Human GECs were treated with 10 μM of DCF-DA in the presence or absence of febuxostat at 37 °C for 30 min. Then, human GECs were washed in phosphate-buffered saline. The green fluorescence from the cells was assessed via fluorescence microscopy (Eclipse TE300, Nikon, Tokyo, Japan). The fluorescence intensity was calculated using the Image-J software (The National Institutes of Health, Bethesda, MD, USA).

### 4.9. Statistical Analysis

Data were presented as means ± standard deviation. Multiple comparisons between groups were performed using one-way analysis of variance (ANOVA) with Bonferroni correction (SPSS version 20.0 software program, IBM Corp., Armonk, NY, USA). Statistical significance was assumed at *p* < 0.05.

## 5. Conclusions

Inhibition of XO by febuxostat was associated with suppression of VEGF/VEGFR1 and VEGFR3 axis and subsequent Akt phosphorylation, which resulted in attenuation of endothelial dysfunction in DKD mice and HG-induced human GECs. The XO inhibition also hindered NOX expression and FoxO3a phosphorylation, resulting in the prevention of oxidative stress. Figure 9 shows the schematic illustration of the proposed molecular mechanism for renoprotective effects of XO inhibition in DKD. In conclusion, XO inhibition protects against DKD by attenuating oxidative stress and endothelial cell dysfunction via the VEGF/VEGFR axis and NOX-FoxO3a-eNOS signaling pathway, and it represents a potential therapeutic target to prevent DKD. Further clinical or experimental research may be needed to fully understand the effects of XO inhibition in DKD.

## Figures and Tables

**Figure 2 ijms-24-03807-f002:**
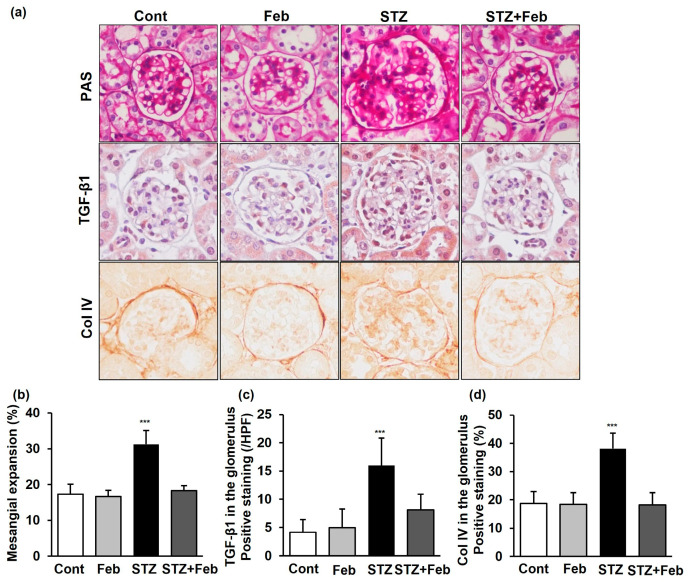
Changes in the glomerular phenotype in STZ-induced diabetic mice with or without febuxostat treatment. (**a**) Representative sections stained with periodic acid–Schiff reagent and representative immunohistochemical staining for TGF-β1 and type IV collagen (original magnification 400×). (**b**–**d**) Quantitative analyses of the results for mesangial fractional area (%), TGF-β1 and type IV collagen (fold) *** *p* < 0.001 vs. other groups.

**Figure 3 ijms-24-03807-f003:**
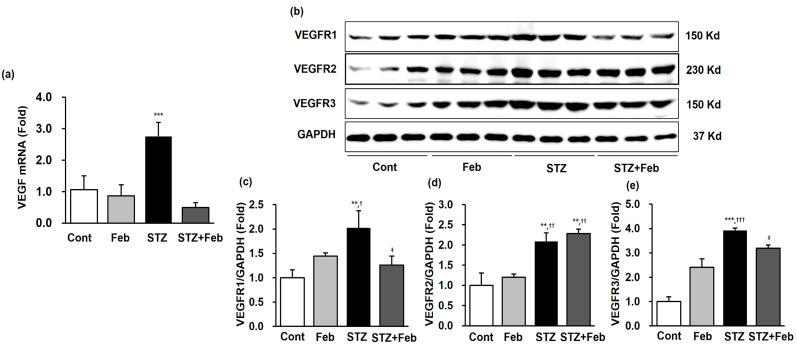
The expression of VEGF mRNA and representative images and quantitative analysis of VEGFR expressions in STZ-induced diabetic mice with or without febuxostat treatment. (**a**) The renal expression levels of VEGF mRNA, *** *p* < 0.001 vs. other groups. (**b**) Representative immunoblots for VEGFR1, VEGFR2, and VEGFR3 expression. (**c**) Quantitative analyses for VEGFR1/GAPDH. (**d**) Quantitative analyses for VEGFR2/GAPDH. (**e**) Quantitative analyses for VEGFR3/GAPDH, ** *p* < 0.01 and *** *p* < 0.001 vs. Cont, ^†^ *p* < 0.05, ^††^
*p* < 0.01 and ^†††^ *p* < 0.001 vs. Feb, and ^‡^ *p* < 0.05 vs. STZ group.

**Figure 4 ijms-24-03807-f004:**
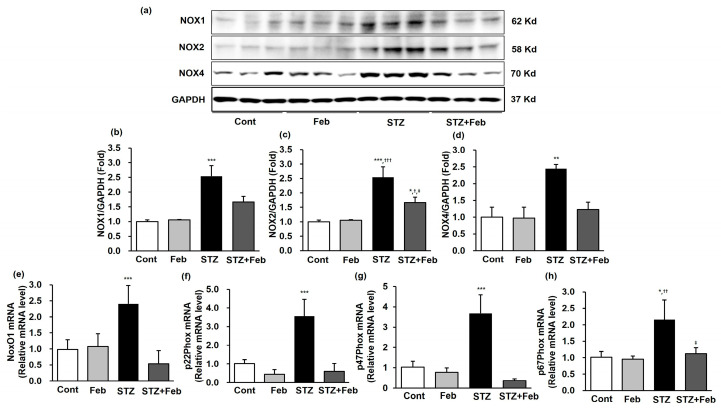
Representative images and quantitative analysis of NADPH oxidase and the mRNA expression of NADPH oxidase subunits in STZ-induced diabetic mice with or without febuxostat treatment. (**a**) Representative immunoblots of NOX1, NOX2 and NOX4 expressions. (**b**) Quantitative analyses for NOX1/GAPDH, *** *p* < 0.001 vs. other groups. (**c**) Quantitative analyses for NOX2/GAPDH, * *p* < 0.05 and *** *p* < 0.001 vs. Cont, ^†^ *p* < 0.05 and ^†††^ *p* < 0.001 vs. Feb and ^‡^
*p* < 0.05 vs. STZ group. (**d**) Quantitative analyses for NOX4/GAPDH, ** *p* < 0.01 vs. other groups. (**e**) NoxO1 mRNA, *** *p* < 0.001 vs. other groups. (**f**) p22phox mRNA, *** *p* < 0.001 vs. other groups. (**g**) p47phox mRNA, *** *p* < 0.001 vs. other groups. (**h**) p67phox mRNA, * *p* < 0.05 vs. Cont, ^††^ *p* < 0.01 vs. Feb and ^‡^
*p* < 0.05 vs. STZ group.

**Figure 5 ijms-24-03807-f005:**
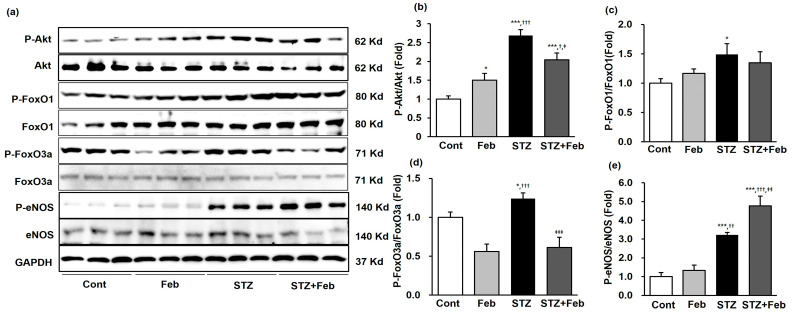
Representative images and quantitative analysis for Akt, FoxOs and eNOS expressions in STZ-induced diabetic mice with or without febuxostat treatment. (**a**) Representative immunoblot showing Akt, FoxOs and eNOS expression levels in mouse kidneys. (**b**) Quantitative analyses for phosphor-Ser^473^ Akt/total Akt, (**c**) Quantitative analyses for phospho-Ser^256^ FoxO1/total-FoxO1. (**d**) Quantitative analyses for phospho-Ser^253^ FoxO3a/total-FoxO3a. (**e**) Quantitative analyses for phospho-Ser^1173^ eNOS/total eNOS, * *p* < 0.05 and *** *p* < 0.001 vs. Cont, ^††^ *p* < 0.01 and ^†††^ *p* < 0.001 vs. Feb and ^‡^ *p* < 0.05, ^‡‡^ *p* < 0.01, and ^‡‡‡^ *p* < 0.001 vs. STZ group.

**Figure 6 ijms-24-03807-f006:**
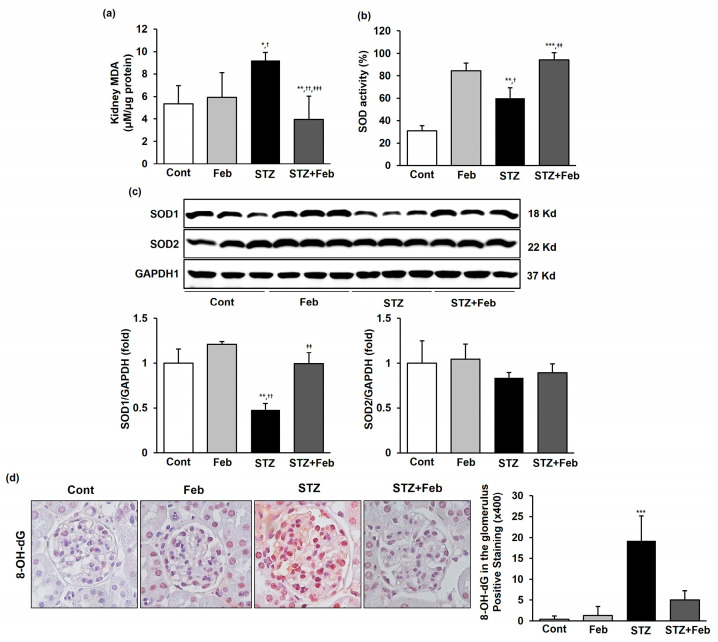
The effects of febuxostat on intrarenal MDA, SOD activity, and immunohistochemical staining for 8-OH-dG in STZ-induced diabetic mice. (**a**) Concentrations of the lipid peroxidation marker MDA in mouse kidneys. * *p* < 0.05 and ** *p* < 0.01 vs. Cont, ^†^ *p* < 0.05 and ^††^ *p* < 0.01 vs. Feb, and ^‡‡‡^ *p* < 0.001 vs. STZ group. (**b**) SOD activity in mouse kidneys. ** *p* < 0.01 and *** *p* < 0.001 vs. Cont, ^†^ *p* < 0.05 vs. Feb, and ^‡‡^ *p* < 0.01 vs. STZ group. (**c**) Representative images and quantitative analyses for SOD1/GAPDH and SOD2/GAPDH, ** *p* < 0.01 vs. Cont, ^††^ *p* < 0.01 vs. Feb and ^‡‡^ *p* < 0.01 vs. STZ group. (**d**) Representative immunohistochemical staining and semiquantitative analysis for 8-OH-dG in mouse kidneys (original magnification 400×). *** *p* < 0.001 vs. other groups.

**Figure 7 ijms-24-03807-f007:**
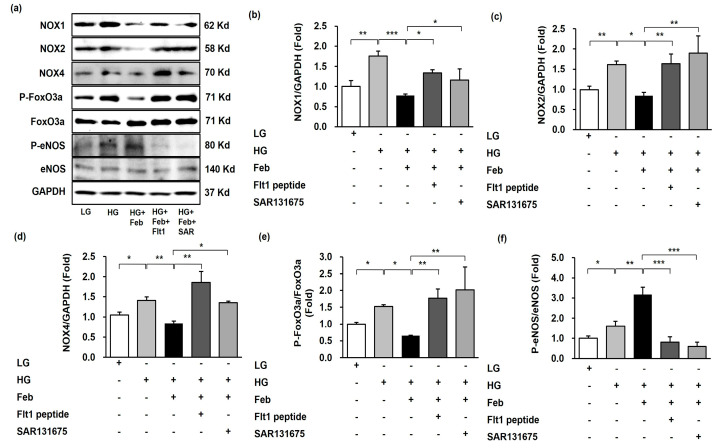
Representative images and quantitative analysis for NOX1, NOX2, NOX4, FoxO3a, and eNOS according to VEGFR1 or VEGFR3 inhibition in HG-treated human GECs with or without febuxostat. (**a**) Representative immunoblot images of NOX1, NOX2, NOX4, FoxO3a and eNOS. (**b**) Quantitative analyses of NOX1/GAPDH. (**c**) Quantitative analyses of NOX2/GAPDH. (**d**) Quantitative analyses of NOX4/GAPDH. (**e**) Quantitative analyses for phospho-Ser^253^ FoxO3a/total FoxO3a. (**f**) Quantitative analyses for phospho-Ser^1173^ eNOS/total eNOS. * *p* < 0.05, ** *p* < 0.01, and *** *p* < 0.001, Fit1: anti-Flt1 peptide, SAR: SAR131675.

**Figure 8 ijms-24-03807-f008:**
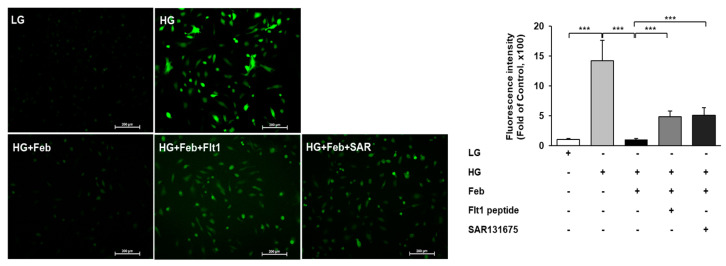
Representative image and quantitative analysis of ROS generation measured by DCF-DA assay according to VEGFR1 or VEGFR3 inhibition in HG-treated human GECs with or without febuxostat. *** *p* < 0.001.

**Figure 9 ijms-24-03807-f009:**
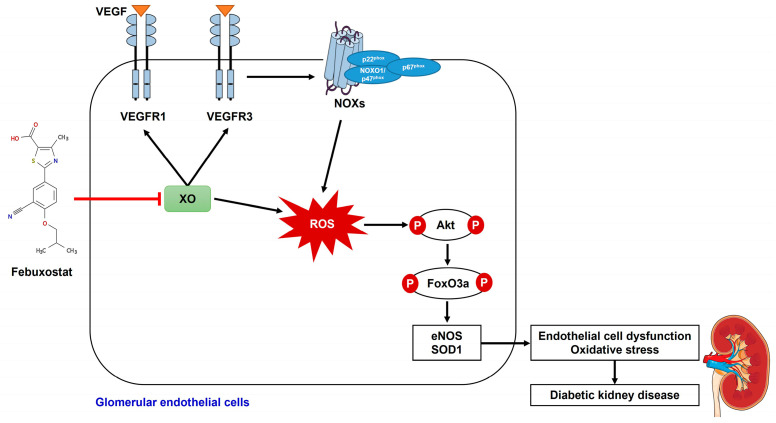
The schematic illustration of the proposed molecular mechanism for renoprotective effects of XO inhibition in DKD.

**Table 2 ijms-24-03807-t002:** Primer sequences for reverse transcription polymerase chain reaction analysis.

Gene	Sequence (Mouse)
*VEGF*	Fwd: CAGGCTGCTGTAACGATGAA Rev: AAATGCTTTCTCCGCTCTGA
*NoxO1*	Fwd: GACATTTGCCTTCTCCGTGT Rev: CGTACCAGTCCTCGACCAGT
*p22Phox*	Fwd: TGGACGTTTCACACAGTGGT Rev: ACCGACAACAGGAAGTGGAG
*p47Phox*	Fwd: ACCTGAAACTGCCCACTGAC Rev: CTGTTCCCGAACTCTTCTCG
*p67Phox*	Fwd: TGGCCTACTTCCAGAGAGGA Rev: CTTCATGTTGGTTGCCAATG
*mGAPDH*	Fwd: TGCAGTGGCAAAGTGGAGATT Rev: CGTGAGTGGAGTCATACTGGAACA

Abbreviations: Fwd—forward; Rev—reverse; GAPDH—glyceraldehyde-3-phosphate dehydrogenase; NoxO1—NADPH oxidase organizer 1; VEGF—vascular endothelial growth factor.

## Data Availability

The data presented in this study are available on request from the corresponding author upon reasonable request.

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
