# Peer review of "Inhibition of Xanthine Oxidase Protects against Diabetic Kidney Disease through the Amelioration of Oxidative Stress via VEGF/VEGFR Axis and NOX-FoxO3a-eNOS Signaling Pathway"

_ijms, 2023, doi:10.3390/ijms24043807_

Round 1

Reviewer 1 Report

In this manuscript (MS), the authors investigated whether XO inhibition exerts renoprotective effects by inhibiting vascular endothelial growth factor (VEGF) and NADPH oxidase (NOX) on diabetic kidney disease (DKD).

There is a list of comments that the authors should take in consideration:

- The manuscript requires proofreading and revision to improve the quality of English. 

- In results, the authors used western blotting data of GADPH for all figures. How then the authors can explian this issue?

- Statistical analysis should be revised by an expert in the field. This is really important.

- Indeed, the authors should strength the Disccussion and conclusion of this study.

- A graphical abstract summarizing the role of XO inhibition in ameliorating oxidative stress and the underlying mechanisms. 

Author Response

Response to Reviewer comments:

Reviewer 1

In this manuscript (MS), the authors investigated whether XO inhibition exerts renoprotective effects by inhibiting vascular endothelial growth factor (VEGF) and NADPH oxidase (NOX) on diabetic kidney disease (DKD). There is a list of comments that the authors should take in consideration:

Response) Thank you for your comments and suggestions that allowed us to greatly improve the quality of the manuscript.

  1. The manuscript requires proofreading and revision to improve the quality of English.

Response) Thank you for your precise comments. Another reviewer also pointed to this problem, and we agree with your comments. We had already proofread this manuscript before submitting your journal, but the proofreading seems to be not enough. We checked the manuscript for revision thoroughly and corrected grammatical errors in the manuscript. In addition, the grammatical errors were minimized by giving proofreading to native speakers and the relevant supporting documents are attached below.

  1. In results, the authors used western blotting data of GADPH for all figures. How then the authors can explain this issue?

Response) Thank you for your precise comments. We replaced the immunoblot of GAPDH for each figure in the revised manuscript.

  1. Statistical analysis should be revised by an expert in the field. This is really important.

Response) We would you like to thank you for your precise comments. We reanalyze multiple comparisons of all experiments using one-way analysis of variance with Bonferroni correction.

  • Line 455: Multiple comparisons between groups were performed using one-way analysis of variance (ANOVA) with Bonferroni correction (SPSS version 20.0 software program, IBM Corp., Armonk, NY, USA).

  1. Indeed, the authors should strength the Discussion and conclusion of this study.

Response) We are very grateful to the reviewer for a detailed review. We included additional comments in the discussion and conclusion of the revised manuscript.

  • Line 277: we revealed the possible mechanisms responsible for the beneficial effect of XO inhibition on endothelial cell damage of DKD caused by oxidative stress in the present study using in vivo and in vitro
  • Line 331: FoxO transcription factors play a crucial role in regulating cellular homeostasis by inducing the expression of genes involved in cellular metabolism and resistance to oxidative stress [43]. Our previous studies revealed that diabetic conditions induced PI3K and Akt, leading to FoxO1 and FoxO3a phosphorylation in type 2 DKD model using db/db mice [44,45]. In addition, we recently reported that XO inhibition attenuated contrast-induced acute kidney injury by modulating oxidative stress through the inhibition of NOX1 and NOX2, while simultaneously enhancing dephosphorylation of FoxO1 and FoxO3a [46]. In this study, NOXs expressions and FoxO1 and FoxO3a phosphorylation increased in STZ-induced DKD mice, whereas XO inhibition suppressed NOXs activation and FoxO3a phosphorylation. Febuxostat-induced FoxO3a dephosphorylation and NOXs suppression were diminished by VEGFR1 inhibitor or VEGFR3 inhibitor in HG-treated human GECs. Our data suggest that targeting the VEGF-NOXs-FoxO3a signaling pathway through the XO inhibition could have beneficial effects, alleviating intracellular oxidative stress in DKD.
  • Line 465: In conclusion, XO inhibition protects against DKD by attenuating oxidative stress and endothelial cell dysfunction via the VEGF/VEGFR axis and NOX-FoxO3a-eNOS signaling pathway, and it represents a potential therapeutic target to prevent DKD. Further clinical or experimental research may be needed to fully understand the effects of XO inhibition in DKD.

  1. A graphical abstract summarizing the role of XO inhibition in ameliorating oxidative stress and the underlying mechanisms.

Response) Thank you for your careful comments. As a reviewer’s suggestion, we added a graphical abstract in the revised manuscript.

  • Line 464: Figure 9 shows the schematic illustration of the proposed mechanism for renoprotective effects of XO inhibition in DKD.

Figure 9. The schematic illustration of the proposed molecular mechanism for renoprotective effects of XO inhibition in DKD.

Reviewer 2 Report

Manuscript Number: ijms-2191092

Journal: International Journal of Molecular Science

Manuscript Title: Inhibition of xanthine oxidase protects against diabetic kidney  disease through the amelioration of oxidative stress via VEGF/VEGFR axis and  NOX-FoxO3a-eNOS signaling pathway

The manuscript is interesting. However, there are still some problems to be solved. Below is my comments.

Section Title

1. L 2: Revise “protects against” to “protected against”.

Section Abstract

2. L 20: Revise “the mRNA levels” to “mRNA levels”.

3. L 23: Revise “In in vitro study” to “In in vitro study”. Please note italics.

Section Introduction

4. L 35: Revise “the molecular mechanisms” to “molecular mechanisms”.

5. L 36: Revise “stress, which results from” to “stress resulted from”.

6. L 49: Revise “mice models” to “mouse models”.

7. L 56-57: Revise “other cells in the glomerulus such as podocytes and mesangial cells” to “other cells (such as podocytes and mesangial cells) in the glomerulus”.

8. L 59: Revise “Models of” to “The models of”.

9. L 63: Revise “diabetic mice models” to “diabetic mouse models”. Pay attention to similar problems.

10. L 64: Revise “the renoprotective” to “renoprotective”.

11. L 65-67: Regarding “a XO inhibitor could attenuate oxidative stress and protect DKD by inhibiting VEGF-NADPH oxidase (NOX) signaling pathway in an animal model of DKD and human GECs”, ““a XO inhibitor could protect DKD”?.  Is protect against DKD right?

12. The authors carried out a series of studies, but the section introduction did not reflect research progress in relevant fields, and it is suggested to supplement relevant contents. 

Section Results

13. L 69: Revise “Effects of” to “The effects of”.

14. L 70: Revise “period are” to period were”.

15. L 71: Revise “decreased and food and” to “decreased, and food and”. Is the change right?

16. L 72-73: Revise “further increased in the STZ and STZ+Feb groups but without significant difference” to “increased but without significant difference in the STZ and STZ+Feb groups compared with the Cont group”.

17. L 73: Revise “The kidney weight/body weight ratio” to “Kidney weight/body weight ratio”.

18. L 73-74: Regarding “The kidney weight/body weight ratio was significantly increased in the STZ group and significantly improved in the STZ+Feb group”, Please rewrite above sentence. Note: 1) Identify comparison results between which groups; 2) Which table or graph results do these descriptions come from? Please pay attention to similar problems.

19. L 76: Revise “with Cont and Feb groups” to “with the Cont group and Feb group, respectively”. Is the change right?

20. L 78: Revise “a significant difference was not found” to “a non-significant difference was found”. Is the change right?

21. L 80: Revise “signifi-cantly” to “significantly”.

22. L 100: Revise “changes and expression” to “changes, and the expression”.

23. L 102: Revise “between the Cont and Feb groups” to “between the Cont group and the Feb group”.

24. L 108-109: Revise “febux-ostat treatment” to febuxostat treatment”. Please pay attention to similar errors.

25. L 110: Revise “collagen are shown” to “collagen”.

26. L 111: Revise “for the mesangial fractional area (%), TGF-β1” to “for mesangial fractional area (%), TGF-β1,”.

27. L 112: Revise “ (fold) are shown” to “ (fold)”.

28. L 113: Revise “Effects of XO inhibition on VEGF and VEGFR expression” to “The effects of XO inhibition on VEGF and VEGFR expression”.

29. L 118: Revise “with Cont and” to “with the Cont and”.

30. L 129: Regarding “n = 6-8 mice/group”, please provide exact data. 6? 7?, or 8?

31. L 130: Revise “Effects of XO inhibition on NOX expressions” to “The effects of XO inhibition on NOX expressions.

32. L 131: Revise “expressions was” to “expressions were”.

33. L 133: Revise “NOX2, and NOX4 expression were” to “NOX2, and NOX4 expressions were”. Please pay attention to similar errors.

34. L 137-138: Regarding “The STZ-induced induction of all the NOX subunits”, “ STZ-induced induction”?

35. L 164: Revise “the FoxO1” to “FoxO1”. Please pay attention to similar problems.

36. L 176: Revise “The malondialdehyde (MDA) levels in the kidneys of mice in the STZ group were” to “Malondialdehyde (MDA) level in the kidneys of mice in the STZ group was”.

37. L 203: Revise “ROS levels” to “ROS level”.

Section Discussion

38. L 226-227: Please rewrite “the dephosphorylation of Akt and FoxO3a and eNOS phosphorylation” to make the contents clearly.

39. L 233: Revise “acid levels are” to “acid levels were”.

40. L 240: Regarding “aggravated glomerular injury”, “aggravated”?

41. L 241-242: It is suggested to cite references to support elevated oxidative stress markers including MDA levels and 8-OH-dG in the kidney tissues”. For example, supplement “Two findings supported our results. The levels of 8-OHdG, MDA, and H2O2 in the Graves’ ophthalmopathy orbital fibroblasts were higher than those in the normal controls (Tsai et al, 2010). MDA and H2O2 increased in common carp liver oxidative damage caused by toxic 4-tert-butylphenol (Cui et al., 2022)” after “8-OH-dG in the kidney tissues”.

Tsai, CC., Wu, SB., Cheng, CY. et al. Increased oxidative DNA damage, lipid peroxidation, and reactive oxygen species in cultured orbital fibroblasts from patients with Graves’ ophthalmopathy: evidence that oxidative stress has a role in this disorder. Eye 24, 1520–1525 (2010).

Jiawen Cui, Qin Zhou, Meijin Yu et al. 4-tert-butylphenol triggers common carp hepatocytes ferroptosis via oxidative stress, iron overload, SLC7A11/GSH/GPX4 axis, and ATF4/HSPA5/GPX4 axis. Ecotoxicology and Environmental Safety 242:113944 (2022). 

42. L 241-242: Revise markers including MDA levels and 8-OH-dG in the kidney tissues to markers including MDA and 8-OH-dG in kidney tissues.

43. L 249: Revise “XOR levels in human plasma are” to “XOR level in human plasma is.

44. L 255: Revise “attenuates glomerular” to “attenuated glomerular”.

45. L 262: Regarding exogenous XO induced VEGF-induced ROS production”,  Does exogenous XO or VEGF induce ROS production? Please rewrite above contents.

46. L 278-279: Revise “showed VEGFR1” to “showed that VEGFR1”.

47. L 308: Revise “This finding indicates” to “These findings indicated.

Section Materials and methods

48. L 311: Revise 4.1. Animal to 4.1. Animals.

49. L 315: Revise “streptozotocin (STZ)-induced DKD (STZ)” to “STZ-induced DKD (STZ)”.

50. L 325-326: Revise “by a pipette” to using a pipette”.

51. L 326-327: Revise “blood harvested” to “blood were harvested”. Is the change right?

52. L 328: Revise “4.2. Assessment of” to “4.2. The assessments of”.

53. L 329: Revise “blood glucose levels were” to “blood glucose level was.

54. L 332: Revise “serum cystatin C levels were assessed using” to “serum cystatin C level was assessed with.

55. Did the authors consider animal ethics?

56. L 340: Revise “in the kidney” to “in kidneys.

57. L 344: Revise “using antibodies” to with antibodies.

58. Please the authors supplement main steps of histopathologic analysis.

59. L 364-365: Revise “for 10 minutes at 4°C to “at 4 °C for 10 minutes.

60. L 365: Revise “Activity of XO and XDH in the kidney was determined using” to The activities of XO and XDH in kidneys were determined with.

61. L 366: Revise “Diego, CA, USA). ” to “Diego, CA, USA), respectively. ”. Is the change right?

62. L 372: Regarding “The concentration of the unknown samples was calculated”, “unknown samples”?

63. Please supplement synthetic information of primers.

64. L 404: Revise Intracellular ROS levels measured” to Intracellular ROS level was measured.

65. L 407: Revise “for 30 minutes at 37°C” to at 37 °C for 30 minutes.

66. L 413: Revise “Data are presented” to “Data were presented”.

67. L 414: Revise “by one-way analysis” to using one-way analysis”.

Section Conclusion

68. L 422: Revise “results highlight” to “results highlighted.

Author Response

Response to Reviewer comments:

Reviewer 2

The manuscript is interesting. However, there are still some problems to be solved. Below are my comments.

Response) Thank you for your precise comments. Another reviewer also pointed to this problem, and we agree with your comments. We had already proofread this manuscript before submitting your journal, but the proofreading seems to be not enough. We checked the manuscript for revision thoroughly and corrected grammatical errors in the manuscript. In addition, the grammatical errors were minimized by giving the proofreading to native speakers and the relevant supporting documents are attached below.

Section Title

  1. L 2: Revise “protects against” to “protected against”.

Response) Thank you for your comments. We corrected the expression according to your comments.

Section Abstract

  1. L 20: Revise “the mRNA levels” to “mRNA levels”.

Response) Thank you for your comment. We corrected the expression according to your comments.

  1. L 23: Revise “In in vitro study” to “In in vitrostudy”. Please note italics.

Response) Thank you for your comments. We corrected the expression according to your comments.

Section Introduction

  1. L 35: Revise “the molecular mechanisms” to “molecular mechanisms”.

Response) Thank you for your comments. We corrected the expression according to your comments.

  1. L 36: Revise “stress, which results from” to “stress resulted from”.

Response) Thank you for your comments. We were recommended other paraphrases from the proofreading, and we changed to it.

  1. L 49: Revise “mice models” to “mouse models”.

Response) Thank you for your comments. We corrected the expression according to your comments.

  1. L 56-57: Revise “other cells in the glomerulus such as podocytes and mesangial cells” to “other cells (such as podocytes and mesangial cells) in the glomerulus”.

Response) Thank you for your comments. We corrected the expression according to your comments.

  1. L 59: Revise “Models of” to “The models of”.

Response) Thank you for your comments. We were recommended other paraphrase from the proofreading, and we changed to it.

  1. L 63: Revise “diabetic mice models” to “diabetic mouse models”. Pay attention to similar problems.

Response) Thank you for your comments. We corrected the expression according to your comments.

  1. L 64: Revise “the renoprotective” to “renoprotective”.

Response) Thank you for your comments. We corrected the expression according to your comments.

  1. L 65-67: Regarding “a XO inhibitor could attenuate oxidative stress and protect DKD by inhibiting VEGF-NADPH oxidase (NOX) signaling pathway in an animal model of DKD and human GECs”, ““a XO inhibitor could protect DKD”?.  Is “protect against DKD” right?

Response) Thank you for your comments. We corrected the expression according to your comments.

  1. The authors carried out a series of studies, but the section introduction did not reflect research progress in relevant fields, and it is suggested to supplement relevant contents. 

Response) We are very grateful to the reviewer for a detailed review. We revised the introduction as you suggested.

  • Line 65: A recent study demonstrated that XO inhibition protected against endometrial hyperplasia through the improvement of oxidative stress and inflammation by improving uterine-reduced glutathione (GSH) and superoxide dismutase (SOD) and inhibiting the ex-pressions of phosphatidylinositol-3-kinase (PI3K), Akt and VEGF [18].

Reference) Mohamed, M.Z.; Baky, M.A.E.; Hassan, O.A.; Mohammed, H.H.; Abdel-Aziz, A.M. PTEN/PI3K/VEGF signaling pathway involved in the protective effect of xanthine oxidase inhibitor febuxostat against endometrial hyperplasia in rats. Hum Exp Toxicol 2020, 39, 1224-1234.

Section Results

  1. L 69: Revise “Effects of” to “The effects of”.

Response) Thank you for your comments. We corrected the expression according to your comments.

  1. L 70: Revise “period are” to “period were”.

Response) Thank you for your comments. The proofreading recommended us to keep “are”.

  1. L 71: Revise “decreased and food and” to “decreased, and food and”. Is the change right?

Response) Thank you for your comments. We were recommended other paraphrase from the proofreading, and we changed to it.

  1. L 72-73: Revise “further increased in the STZ and STZ+Feb groups but without significant difference” to “increased but without significant difference in the STZ and STZ+Feb groups compared with the Cont group”.

Response) Thank you for your comments. We corrected the expression according to your comments.

  1. L 73: Revise “The kidney weight/body weight ratio” to “Kidney weight/body weight ratio”.

Response) Thank you for your comments. We corrected the expression according to your comments.

  1. L 73-74: Regarding “The kidney weight/body weight ratio was significantly increased in the STZ group and significantly improved in the STZ+Feb group”, Please rewrite above sentence. Note: 1) Identify comparison results between which groups; 2) Which table or graph results do these descriptions come from? Please pay attention to similar problems.

Response) Thank you for your comments. We corrected the expression according to your comments.

  • Line 78: Kidney weight/body weight ratio significantly increased in the STZ group compared with the Cont and Feb groups and significantly decreased in the STZ+Feb group (Table 1, p < 0.01).

  1. L 76: Revise “with Cont and Feb groups” to “with the Cont group and Feb group, respectively”. Is the change right?

Response) Thank you for your comments. We corrected the expression according to your comments.

  1. L 78: Revise “a significant difference was not found” to “a non-significant difference was found”. Is the change right?

Response) Thank you for your comments. We were recommended other paraphrase from the proofreading, and we changed to it.

  1. L 80: Revise “signifi-cantly” to “significantly”.

Response) Thank you for your comments. We corrected the expression according to your comments.

  1. L 100: Revise “changes and expression” to “changes, and the expression”.

Response) Thank you for your comments. We corrected the expression according to your comments.

  1. L 102: Revise “between the Cont and Feb groups” to “between the Cont group and the Feb group”.

Response) Thank you for your comments. We were recommended to remove “the” from the proofreading.

  1. L 108-109: Revise “febux-ostat treatment” to “febuxostat treatment”. Please pay attention to similar errors.

Response) Thank you for your comments. We corrected the expression according to your comments.

  1. L 110: Revise “collagen are shown” to “collagen”.

Response) Thank you for your comments. We corrected the expression according to your comments.

  1. L 111: Revise “for the mesangial fractional area (%), TGF-β1” to “for mesangial fractional area (%), TGF-β1,”.

Response) Thank you for your comments. We corrected the expression according to your comments.

  1. L 112: Revise “ (fold) are shown” to “ (fold)”.

Response) Thank you for your comments. We corrected the expression according to your comments.

  1. L 113: Revise “Effects of XO inhibition on VEGF and VEGFR expression” to “The effects of XO inhibition on VEGF and VEGFR expression”.

Response) Thank you for your comments. We corrected the expression according to your comments.

  1. L 118: Revise “with Cont and” to “with the Cont and”.

Response) Thank you for your comments. We corrected the expression according to your comments.

  1. L 129: Regarding “n= 6-8 mice/group”, please provide exact data. “6”? “7”?, or “8”?

Response) We are very grateful to the reviewer for a detailed review. We described the number of mice in each group used this study in the method section.

  • Line 348: Mice were randomly assigned into the following groups: control (Cont, n = 6), febuxo-stat-treated (Feb, n = 6), streptozotocin (STZ)-induced DKD (STZ, n = 8), and STZ-induced DKD treated with febuxostat (STZ+Feb, n = 8).

  1. L 130: Revise “Effects of XO inhibition on NOX expressions” to “The effects of XO inhibition on NOX expressions”.

Response) Thank you for your comments. We corrected the expression according to your comments.

  1. L 131: Revise “expressions was” to “expressions were”.

Response) Thank you for your comments. We corrected the expression according to your comments.

  1. L 133: Revise “NOX2, and NOX4 expression were” to “NOX2, and NOX4 expressions were”. Please pay attention to similar errors.

Response) Thank you for your comments. We corrected the expression according to your comments.

  1. L 137-138: Regarding “The STZ-induced induction of all the NOX subunits”, “ STZ-induced induction”?

Response) Thank you for your comments. We were recommended other paraphrase from the proofreading, and we changed to it.

  1. L 164: Revise “the FoxO1” to “FoxO1”. Please pay attention to similar problems.

Response) Thank you for your comments. We corrected the expression according to your comments.

  1. L 176: Revise “The malondialdehyde (MDA) levels in the kidneys of mice in the STZ group were” to “Malondialdehyde (MDA) level in the kidneys of mice in the STZ group was”.

Response) Thank you for your comments. We corrected the expression according to your comments.

  1. L 203: Revise “ROS levels” to “ROS level”.

Response) Thank you for your comments. We corrected the expression according to your comments.

Section Discussion

  1. L 226-227: Please rewrite “the dephosphorylation of Akt and FoxO3a and eNOS phosphorylation” to make the contents clearly.

Response) Thank you for your comments. We corrected the expressions according to your comments.

  • Line 242: The protective effects of febuxostat for DKD were attributed to the dephosphorylation of Akt and FoxO3a and the enhancement of eNOS phosphorylation, which reversed renal oxidative stress.

  1. L 233: Revise “acid levels are” to “acid levels were”.

Response) Thank you for your comments. We corrected the expression according to your comments.

  1. L 240: Regarding “aggravated glomerular injury”, “aggravated”?

Response) Thank you for your comments. We corrected other paraphrase according to your comments.

  • Line 255: these findings were accompanied with the increase in albuminuria and serum cystatin C levels and the aggravation of glomerular injury.

  1. L 241-242: It is suggested to cite references to support “elevated oxidative stress markers including MDA levels and 8-OH-dG in the kidney tissues”. For example, supplement “Two findings supported our results. The levels of 8-OHdG, MDA, and H2Oin the Graves’ ophthalmopathy orbital fibroblasts were higher than those in the normal controls (Tsai et al, 2010). MDA and H2O2increased in common carp liver oxidative damage caused by toxic 4-tert-butylphenol (Cui et al., 2022)” after “8-OH-dG in the kidney tissues”.

Tsai, CC., Wu, SB., Cheng, CY. et al. Increased oxidative DNA damage, lipid peroxidation, and reactive oxygen species in cultured orbital fibroblasts from patients with Graves’ ophthalmopathy: evidence that oxidative stress has a role in this disorder. Eye 24, 1520–1525 (2010).

Jiawen Cui, Qin Zhou, Meijin Yu et al. 4-tert-butylphenol triggers common carp hepatocytes ferroptosis via oxidative stress, iron overload, SLC7A11/GSH/GPX4 axis, and ATF4/HSPA5/GPX4 axis. Ecotoxicology and Environmental Safety 242:113944 (2022). 

Response) Thank you for your comments. We corrected the expressions according to your comments.

  1. L 241-242: Revise “markers including MDA levels and 8-OH-dG in the kidney tissues” to “markers including MDA and 8-OH-dG in kidney tissues”.

Response) Thank you for your comments. We corrected the expression according to your comments.

  1. L 249: Revise “XOR levels in human plasma are” to “XOR level in human plasma is”.

Response) Thank you for your comments. We corrected the expression according to your comments.

  1. L 255: Revise “attenuates glomerular” to “attenuated glomerular”.

Response) Thank you for your comments. We were recommended to change from “d” to “s” from the proofreading.

  1. L 262: Regarding “exogenous XO induced VEGF-induced ROS production”, Does “exogenous XO” or “VEGF” induce ROS production? Please rewrite above contents.

Response) Thank you for your comments. We rewrote this sentence according to your comments.

  1. L 278-279: Revise “showed VEGFR1” to “showed that VEGFR1”.

Response) Thank you for your comments. We corrected the expression according to your comments.

  1. L 308: Revise “This finding indicates” to “These findings indicated”.

Response) Thank you for your comments. We were recommended to change from “indicated” to “indicate” from the proofreading.

Section Materials and methods

  1. L 311: Revise “4.1. Animal” to “4.1. Animals”.

Response) Thank you for your comments. We corrected the expression according to your comments.

  1. L 315: Revise “streptozotocin (STZ)-induced DKD (STZ)” to “STZ-induced DKD (STZ)”.

Response) Thank you for your comments. We corrected the expression according to your comments.

  1. L 325-326: Revise “by a pipette” to “using a pipette”.

Response) Thank you for your comments. We corrected the expression according to your comments.

  1. L 326-327: Revise “blood harvested” to “blood were harvested”. Is the change right?

Response) Thank you for your comments. We corrected the expression according to your comments.

  1. L 328: Revise “4.2. Assessment of” to “4.2. The assessments of”.

Response) Thank you for your comments. We corrected the expression according to your comments.

  1. L 329: Revise “blood glucose levels were” to “blood glucose level was”.

Response) Thank you for your comments. We were recommended to change from “blood glucose level was” to “blood glucose levels were” from the proofreading.

  1. L 332: Revise “serum cystatin C levels were assessed using” to “serum cystatin C level was assessed with”.

Response) Thank you for your comments. We corrected the expression according to your comments.

  1. Did the authors consider animal ethics?

Response) Thank you for your comments. We already included the sentence of animal ethics and the approval number of animal IRB at the end of the manuscript.

  1. L 340: Revise “in the kidney” to “in kidneys”.

Response) Thank you for your comments. We corrected the expression according to your comments.

  1. L 344: Revise “using antibodies” to “with antibodies”.

Response) Thank you for your comments. We corrected the expression according to your comments.

  1. Please the authors supplement main steps of histopathologic analysis.

Response) Thank you for your careful review. As a reviewer’s suggestion, we supplemented the process of histopathologic analysis in the revised manuscript.

  • Line 378: To assess the severity of the mesangial matrix expansion, the fraction mesangial area was calculated as the proportion of the area of mesangial matrix to the total area of glomerulus and presented in percentage by determining the color intensity per glomerulus using Image-J software (the National Institutes of Health, Bethesda, MD, USA) [47].

Reference) Rangan, G.K.; Tesch, G.H. Quantification of renal pathology by image analysis. Nephrology (Carlton) 2007, 12, 553-558.

  1. L 364-365: Revise “for 10 minutes at 4°C” to “at 4 °C for 10 minutes”.

Response) Thank you for your comments. We corrected the expression according to your comments.

  1. L 365: Revise “Activity of XO and XDH in the kidney was determined using” to “The activities of XO and XDH in kidneys were determined with”.

Response) Thank you for your comments. We corrected the expression according to your comments.

  1. L 366: Revise “Diego, CA, USA). ” to “Diego, CA, USA), respectively. ”. Is the change right?

Response) Thank you for your comments. We corrected the expression according to your comments.

  1. L 372: Regarding “The concentration of the unknown samples was calculated”, “unknown samples”?

Response) Thank you for your comments. We corrected the expression according to your comments.

  • Line 413: The concentration of each sample was calculated by comparing the optimal density (OD) of each sample at a wavelength of 450 nm with an OD standard curve.

  1. Please supplement synthetic information of primers.

Response) Thank you for your careful review. As a reviewer’s suggestion, we supplemented synthetic information of primers in the revised manuscript.

  • Line 436: the primers (GenoTech Corporation, Daejeon, Korea) were then amplified using a Power SYBR® Green polymerase chain reaction (PCR) Master Mix (Applied Biosystems, Foster City, CA, USA) with gene-specific primer pairs (Table 2).

  1. L 404: Revise “Intracellular ROS levels measured” to “Intracellular ROS level was measured”.

Response) Thank you for your comments. We corrected the expression according to your comments.

  1. L 407: Revise “for 30 minutes at 37°C” to “at 37 °C for 30 minutes”.

Response) Thank you for your comments. We corrected the expression according to your comments.

  1. L 413: Revise “Data are presented” to “Data were presented”.

Response) Thank you for your comments. We corrected the expression according to your comments.

  1. L 414: Revise “by one-way analysis” to “using one-way analysis”.

Response) Thank you for your comments. We corrected the expression according to your comments.

Section Conclusion

  1. L 422: Revise “results highlight” to “results highlighted”.

Response) Thank you for your comments. We corrected other paraphrase according to your comments.

Reviewer 3 Report

Authors described the role of Xanthine oxidase (XO) and inhibition of XO preserved the renal function in the setting of diabetic kidney disease. Authors investigated the action mechanism of XO inhibitor Febuxostat which scavenge the oxidative stress by inhibiting the VEGF/VEGFR/NADPH oxidase signaling pathway in a mouse model of STZ induced type 1 diabetes and in vitro cell culture model under elevated glucose conditions. The experimental design is well established and in vivo and in vitro results were convincing. However, authors need to address the concerns before the current manuscript accepted for publication in IJMS.

Major concerns:

1) Figure 1 a, results indicating the decreasing of blood glucose levels in STZ mice after treatment with Febuxostat. Therefore, the improved kidney function can be systemic rather specific to kidney? The Febuxostat treatment have any off-target effects by improving the insulin sensitivity and signaling pathway?

2) Figure 2a, showing TGFba increased in glomeruli of STZ mice and decreased significantly upon Febuxostat treatment. Is Febuxostat also inhibiting TGFb-Smad signaling? I do not see any explanation or description in discussion regarding TGFb1-smad signaling in relation to VEGF/VEGFR signaling?

3) Based on several studies, it has been shown that the higher oxidative stress can be seen tubules under DKD condition as these ROS generated from mitochondrial dysfunction. authors showed the renal injury in glomeruli and improved the pathological changes upon Febuxostat treatment. However, did authors see any changes in tubular injury in STZ kidney compared to controls? If yes, is there any improvement in tubular injury markers (for instance KIM1) and interstitial fibrosis upon Febuxostat treatment?

4) Figure 6b, indicating the anti-oxidative enzyme SOD activity. I realized that, SOD activity is increased in STZ model compared to controls and further increased by Febuxostat treatment. How does authors explain the increasing SOD activity which is anti-oxidative action? Need to explain. Also, I would suggest authors can perform the westernblot for SOD?  

5) In methods section 4.4, In vitro experiments: Human GEC culture conditions need to indicate clearly. How long (hours) the cells were cultured in High glucose (HG) medium? How long the Febuxostat was added to cells? Like intervention or prevention?

6) This is excellent study and I would suggest authors can provide the summary of schematic diagram representing the signaling mechanism under DKD and blockage of specific signaling by Febuxostat. This will allow the readers to understand the summary of entire study easily and greatly improve the impact of manuscript.

Author Response

Response to Reviewer comments:

Reviewer 3

Authors described the role of Xanthine oxidase (XO) and inhibition of XO preserved the renal function in the setting of diabetic kidney disease. Authors investigated the action mechanism of XO inhibitor Febuxostat which scavenge the oxidative stress by inhibiting the VEGF/VEGFR/NADPH oxidase signaling pathway in a mouse model of STZ induced type 1 diabetes and in vitro cell culture model under elevated glucose conditions. The experimental design is well established and in vivo and in vitro results were convincing. However, authors need to address the concerns before the current manuscript accepted for publication in IJMS.

Response) Thank you for your comments and suggestions that allowed us to greatly improve the quality of the manuscript.

Major concerns:

1) Figure 1a, results indicating the decreasing of blood glucose levels in STZ mice after treatment with Febuxostat. Therefore, the improved kidney function can be systemic rather specific to kidney? The Febuxostat treatment have any off-target effects by improving the insulin sensitivity and signaling pathway?

Response) We are very grateful to the reviewer for a detailed review. The purpose of our study is not to show the glucose-lowering effect of febuxostat treatment in diabetic kidney disease. Therefore, we did not measure the insulin sensitivity and related signaling pathway. In Table 1, there was no significant difference in HbA1c level between the STZ group and the STZ+Feb group. HbA1c level is a more reliable marker for evaluation of long-term glycemic control than fasting glucose level. Furthermore, we reanalyzed fasting glucose level and found a non-significant difference between the STZ group and the STZ+Feb group using ANOVA with Bonferroni correction in the revised manuscript. Therefore, we decided that febuxostat did not affect glucose-lowering action in diabetic kidney disease.

  • Line 83: Therefore, febuxostat treatment per se did not appear to have any effect on lowering blood glucose (Figure 1 and Table 1).

2) Figure 2a, showing TGFb1 increased in glomeruli of STZ mice and decreased significantly upon Febuxostat treatment. Is Febuxostat also inhibiting TGFb-Smad signaling? I do not see any explanation or description in discussion regarding TGFb1-smad signaling in relation to VEGF/VEGFR signaling?

Response) We are very grateful to the reviewer for a detailed review. TGF-β1 has been implicated as a key mediator of diabetic kidney disease. Therefore, we used TGF-β1 as a molecular biomarker for the development of diabetic kidney disease in STZ-induced type 1 diabetic mouse model in this study. The discussion of TGF-β1-smad signaling in relation to VEGF/VEGFR signaling is not the focus of our study.

3) Based on several studies, it has been shown that the higher oxidative stress can be seen tubules under DKD condition as these ROS generated from mitochondrial dysfunction. authors showed the renal injury in glomeruli and improved the pathological changes upon Febuxostat treatment. However, did authors see any changes in tubular injury in STZ kidney compared to controls? If yes, is there any improvement in tubular injury markers (for instance KIM1) and interstitial fibrosis upon Febuxostat treatment?

Response) We are very grateful to the reviewer for a detailed review. Pronounced up-regulation of VEGF is found in the glomerular endothelial cells in diabetic kidney disease, and VEGF is critical to the development and progression of diabetic kidney disease. As we mentioned in the discussion, circulating XO mainly binds endothelial cells and affects endothelial dysfunction. Therefore, the protective effects of XO inhibition for tubular injury of diabetic kidney disease are not the focus of our study. As a reviewer’s suggestion, we will plan to conduct further experimental research on the protective effects of XO inhibition for tubular injury of diabetic kidney disease in the next study.

4) Figure 6b, indicating the anti-oxidative enzyme SOD activity. I realized that, SOD activity is increased in STZ model compared to controls and further increased by Febuxostat treatment. How does authors explain the increasing SOD activity which is anti-oxidative action? Need to explain. Also, I would suggest authors can perform the westernblot for SOD? 

Response) We would like to thank the reviewer’s comments, and deeply agree with your suggestions. We performed immunoblots of SOD1 and SOD2 and added the results in the revised manuscript.

  • Line 184: In addition, SOD1 expression significantly decreased in the STZ group but enhanced in the STZ+Feb group (Figure 6c, p < 0.01). In contrast, SOD2 expression did not change among the experimental groups.

5) In methods section 4.4, In vitro experiments: Human GEC culture conditions need to indicate clearly. How long (hours) the cells were cultured in High glucose (HG) medium? How long the Febuxostat was added to cells? Like intervention or prevention?

Response) Thank you for your precise and careful comments. As a reviewer’s suggestion, we added the method of in vitro study in the revised manuscript.

  • Line 395: The Feb group was treated with 0.5 µM of febuxostat dissolved in dimethyl sulfoxide (DMSO) and the Cont group with DMSO for 48 hours after 24 hours of HG or LG condition.

6) This is excellent study and I would suggest authors can provide the summary of schematic diagram representing the signaling mechanism under DKD and blockage of specific signaling by Febuxostat. This will allow the readers to understand the summary of entire study easily and greatly improve the impact of manuscript.

Response) Thank you for your precise comments. As a reviewer’s suggestion, we added a graphical abstract in the revised manuscript.

  • Line 464: Figure 9 shows the schematic illustration of the proposed molecular mechanism for renoprotective effects of XO inhibition in DKD. In conclusion, XO inhibition protects against DKD by attenuating oxidative stress and endothelial cell dysfunction via the VEGF/VEGFR axis and NOX-FoxO3a-eNOS signaling pathway, and it represents a potential therapeutic target to prevent DKD. Further clinical or experimental research may be needed to fully understand the effects of XO inhibition in DKD.

Figure 9. The schematic illustration of the proposed molecular mechanism for renoprotective effects of XO inhibition in DKD.

Reviewer 4 Report

Please  see attache file for minor corrections.

Author Response

Response to Reviewer comments:

Reviewer 4

Response) Thank you for your comments and suggestions that allowed us to greatly improve the quality of the manuscript.

1) Line 52: This part of the sentence does not make sense. Please reformulate.

Response) Thank you for your precise comments. We reformulated the sentence in the revised manuscript.

  • Line 52: glomerular endothelial cell dysfunction plays a crucial role in the development and progression of DKD [10].

2) Line 77 and line 96: You see too often “however”.

Response) Thank you for your precise comments. We changed from the word “however” to other words in the revised manuscript. 

  • Line 33: Nevertheless, the need for innovative treatments to prevent or slow the progression of DKD remains unmet.
  • Line 80: Fasting blood glucose and HbA1c levels in the STZ and STZ+Feb groups significantly in-creased compared with the Cont and Feb groups but had a non-significant difference be-tween the two groups, respectively.
  • Line 103: The serum uric acid level did not change in the STZ group, while the increase in uric acid was significantly suppressed in the STZ+Feb group (p < 0.05, Figure 1d).

3) Line 152: Starting with figure 5, there is a delay in figures and their explanation.

Response) Thank you for your precise comments. As a reviewer’s suggestion, we rearranged figures and texts starting with figure 5 in the revised manuscript.

4) Conclusion: The conclusions need expanding.

Response) Thank you for your precise comments. As a reviewer’s suggestion, we added a graphical abstract in the revised manuscript.

  • Line 464: Figure 9 shows the schematic illustration of the proposed mechanism for renoprotective effects of XO inhibition in DKD. In conclusion, XO inhibition protects against DKD by at-tenuating oxidative stress and endothelial cell dysfunction via the VEGF/VEGFR axis and NOX-FoxO3a-eNOS signaling pathway, and it represents a potential therapeutic target to prevent DKD. Further clinical or experimental research may be needed to fully understand the effects of XO inhibition in DKD.

Figure 9. The schematic illustration of the proposed molecular mechanism for renoprotective effects of XO inhibition in DKD.

Round 2

Reviewer 1 Report

The MS. can be accepted.

Reviewer 2 Report

The revised manuscript can be published.

Reviewer 3 Report

Thank you for the responses. All the comments were addressed and changes were made in revised manuscript according to reviewer’s suggestions and concerns. The current version can be accepted for publication in IJMAS. Congratulations to authors.